# Oral Care Recommendation for Cystic Fibrosis Patients–Recommendation for Dentists

**DOI:** 10.3390/jcm11102756

**Published:** 2022-05-13

**Authors:** Tamara Pawlaczyk-Kamieńska, Maria Borysewicz-Lewicka, Halina Batura-Gabryel, Szczepan Cofta

**Affiliations:** 1Department of Risk Group Dentistry, Chair of Pediatric Dentistry, Poznan University of Medical Sciences, 61-701 Poznan, Poland; klstomdz@ump.edu.pl; 2Department of Pulmonology, Allergology and Respiratory Oncology, Poznan University of Medical Sciences, 60-569 Poznan, Poland; hagabryel@ump.edu.pl (H.B.-G.); scofta@ump.edu.pl (S.C.)

**Keywords:** cystic fibrosis, oral health needs, oral care guidelines

## Abstract

Cystic fibrosis (CF) is a genetic disease that is caused by a defect in the gene coding for the transmembrane cystic fibrosis transmembrane conductance regulator (CFTR). Research papers published so far point out that despite the numerous dental treatment needs of CF patients, there are no oral care guidelines for this group of patients. The aim of the article is to propose standards of dental prophylactic and therapeutic procedures for CF patients in different age groups. Regardless of the CF patient’s age, dental check-ups should be scheduled at least every 6 months. However, taking into account the actual condition of the individual CF patients, therapeutic visits may be scheduled for earlier dates, to provide well-fitting treatment, considering the level of risk of oral diseases. The described management standards may be helpful and may improve the quality of dental care provided to CF patients.

## 1. Introduction

Cystic fibrosis (CF) is a rare monogenic genetic disease that is inherited in an autosomal recessive manner. It is caused by a defect in the gene coding for the transmembrane cystic fibrosis transmembrane conductance regulator (CFTR), which is responsible for the formation of a chloride channel in the epithelial cells’ membranes. This channel allows not only for the transport of chloride (Cl^−^) ions but also of bicarbonate (HCO_3_^−^), sodium (Na^+^), potassium (K^+^) and calcium (Ca^2+^) ions, as well as water (H_2_O) through the cell membrane. The abnormal function of chloride channels contributes to the faulty distribution of ions between the two sides of the membrane, which results in many clinical symptoms. The widespread location of chloride channels in the human body explains why the disease shows with a polymorphism of symptoms [1,2].

CF was first described and identified as a separate entity in 1938 by Dorothy Andersen, although some of its symptoms, such as the “salty kiss”, had been known for longer. The breakthrough in cystic fibrosis research took place in 1989 when genetic studies allowed for the identification of the CFTR gene [3]. At present, intensive studies on optimal symptomatic treatment are underway, and large-scale research worldwide offers hope for a causal treatment in the coming years. Meanwhile, thanks to screening tests in newborns which enable early diagnoses of the disease already in the asymptomatic period, as well as the advances in conventional treatment, the life spans of these patients have been significantly extended [4]. Therefore, it seems necessary to pay attention to their dental needs.

## 2. CF Patients’ Oral Health Needs

Previous studies on the oral health of CF patients mainly covered people under 18 years of age [5,6,7,8,9]. As shown by the latest publications [8], the problems that CF patients are likely to face include reduced amounts of saliva and—in extreme cases—xerostomia. Moreover, in CF patients, compared to their healthy peers, a lower pH of saliva and its increased density are noted. Their pathomechanism has not yet been fully explained. It is known, however, that during primary saliva (produced in the secretory acini) flow through the excretory ducts, the saliva ionic composition is significantly modified: Na^+^ and Cl^−^ are absorbed and HCO_3_^−^ and K^+^ are secreted into their lumen. In this way, secondary saliva is formed [10]. The element necessary for the proper functioning of striated ducts cells is a chloride channel (located in the luminal side of the cell membrane of the acinar cells and the apical part of cylindrical cells lining the striated ducts), which is responsible for the transport of ions and water through the cell membrane [10,11,12]. Depending on the type of CFTR gene mutation, a chloride channel may function abnormally or may be completely dysfunctional [12,13,14,15,16]. This may bring about various degrees of hyposalivation and changes in the saliva physicobiochemical properties (reduced pH, different qualitative composition, increased density) [12,15,16]. However, it should be emphasized that hyposalivation in this group of patients may also result from their comorbidities, vitamin deficiencies or be a side effect of pharmacotherapy. However, regardless of the disorder pathomechanism, a reduced saliva secretion can lead to many unpleasant and troublesome effects, such as inflammation of the mouth, burning tongue, disturbed taste perception, difficulty eating food, a tendency to ulcer formation or fungal infections. Moreover, not enough saliva—which promotes bacterial biofilm stagnation—and its missing neutralizing effect on the acids produced in the oral cavity, significantly increase the risk of oral infections and diseases (caries, periodontitis). A change in the quantitative and qualitative compositions of saliva affects the ecosystem; it also affects the qualitative and quantitative composition of oral microorganisms, which may favor the initiation and progression of pathological lesions [8]. Clinical symptoms resulting from these changes can significantly reduce the comfort and quality of life of CF patients.

Another disorder observed in CF patients is enamel defect. The pathomechanism of this anomaly has not yet been explained. According to some reports, they may result from the genetic background of the disease [17,18,19,20]. The chloride channel is located in specialized epithelial cells, including ameloblasts [17,18,21,22]. At the present stage of knowledge, CFTR is considered to be one of the so-far-identified proteins stabilizing the pH inside ameloblasts and regulating the transport of Ca_2_^+^ through the cell membrane to its inside [18,21,22]. The H^+^ acidifying the environment is a by-product of the apatite crystal’s formation in the enamel maturation. These protons should be neutralized in order to enable the crystal’s growth to continue [18]. CFTRs basic role is to regulate the Cl^−^/HCO_3_^−^ exchange through cell membranes, thus neutralizing H^+^ and ensuring a neutral pH, necessary for the transport of Ca_2_^+^ and proper mineralization [18,21,22].

The review of the literature dealing with developmental enamel defects in primary dentition did not show a statistically significant difference between CF children and their healthy peers [5]. However, the data on permanent dentition in this group of patients are inconclusive and report a similar or significantly higher prevalence of enamel defects, compared to the control group [5,9]. Research by Pawlaczyk-Kamieńska et al. [9] on oral health in adult CF patients revealed not only a significantly higher prevalence of permanent tooth enamel defects than in the control group but also a higher percentage of teeth with abnormalities. Moreover, in the CF group, the lesions were found on all teeth groups and in the control group only the incisors and first molars were affected.

Available studies on the condition of teeth in CF patients display no significant differences or statistically significantly lower intensity of caries in adolescent patients compared to the control group. The studies on the intensity of caries in adult patients give ambiguous results as well [5]. No statistically significant difference in caries intensity between the study group (aged >19 years; the number of these patients was not given) and the control group was reported by Matrens et al. (2001) [23]. On the other hand, a statistically significantly lower caries intensity was demonstrated by Aps et al. (2001) [24] who studied two 10-people groups (the studied group and the control one) with the participants aged more than 20 years. Pawlaczyk-Kamieńska et al. reported a significantly higher intensity of caries among adult patients [9]. Moreover, in CF patients, compared to the healthy individuals, the authors noted a significantly higher average number of teeth with active caries and a higher average number of teeth removed due to caries, with a similar number of teeth where caries was successfully treated in both groups.

The results of the studies on the accumulation of oral bacterial biofilm showed no significant differences between CF patients and their healthy peers. People under 18 years of age displayed no difference in the rate of gingival bleeding as well. However, gingival bleeding in adult CF patients was statistically less frequent than in the control group [5,8]. It should be noted that gingivitis is assumed to arise when interactions between plaque microorganisms and the host’s immune system cells occur [25]. According to the “specific plaque” theory, the onset and subsequent progression of gingivitis are triggered not so much by the amount of plaque, but primarily by its composition and by its microbiological pathogenicity in particular. In adult CF patients, no correlation between plaque thickness and gums condition may result from frequent long-term pharmacotherapy (also with inhalations), which probably reduces the biofilm pathogenic potential [6,7]. Presumably, the patients develop an individual and specific, pharmacologically-determined, ecological balance of the biofilm, which, however, may be disturbed, e.g., by lowering the host’s resistance to damaging factors. Such a situation, especially with the saliva profile of CF patients, may lead to an unexpected and sudden occurrence of bacterial and/or fungal inflammatory changes in the oral cavity.

Research papers published so far point out that despite numerous dental treatment needs of CF patients as specified in the literature, no therapeutic protocol has been developed. It seems necessary not only to ensure procedures to maintain healthy dentition and periodontium but also to prevent auto-infections these patients are at risk of. Newborn screening tests that enable an early diagnosis of cystic fibrosis and the classification of patients for early general medical treatment, allow for the introduction of targeted dental measures as well. They will suit the real needs and health of CF patients and will take into account the risks resulting from the disease itself, its course, systemic and inhaled pharmacotherapy, as well as the need to use a high-calorie diet, but at the same time cariogenic.

## 3. Objectives

The aim of the article is to propose standards of dental prophylactic and therapeutic procedures for CF patients.

## 4. Recommendations

According to the current state of knowledge, treatments aimed at preventing oral diseases should take into account the actual needs of an individual patient, systemic health, medications taken, dietary and hygienic habits, as well as the risk of oral diseases. Etiological factors of oral diseases pose a threat to CF patients throughout their lives, so oral health should be regularly monitored. The dentist is required to thoroughly diagnose a patient and prepare an individualized preventive and therapeutic plan, which should be adjusted to the changing needs of a CF patient and their current state of health. As there is a multitude of factors influencing oral health, the dates of dental visits should be agreed upon individually and they should be frequent as to fit the CF patients’ needs which arise from the effectiveness of removing bacterial plaque, oral diseases’ activity, risk of caries or its progression and risk of periodontal or mucosal diseases.

With insufficient oral hygiene, in order to assess and possibly correct home dental care, the next visit (the so-called progress control visit) should take place soon, even as early as after 2 weeks [26,27]. The number of such visits will depend on a given CF patient’s cooperation, their physical fitness and well-being related to the course of the disease, as well as oral hygiene results obtained. With proper oral hygiene, good dietary habits and appropriate fluoride supplementation, it is conditionally recommended to make periodic check-ups at least every 6 months [26,27]. Recommended dental procedures for individual age groups of CF patients are presented in Table 1.

### 4.1. Newborns

In order to maintain oral health in infants diagnosed with cystic fibrosis through screening tests, caries prevention measures should be started as early as possible. It is important to inform the parents regarding possible abnormalities that may occur in the child’s oral cavity due to the underlying disease and its pharmacotherapy. In addition, there is a strong recommendation to give them instructions on how to carry out hygienic procedures, first for the toothless and then the toothed mouth of an infant. Parents should be made aware that their oral cavity is a reservoir of microorganisms that can be transferred to a child’s mouth, e.g., through kisses or a teaspoon [28] and so, in order to break the chain of infection, their mouths also need to be sanitized.

Oral health largely depends on the diet used. From the point of view of oral diseases etiology, especially caries etiology, carbohydrates lead to the development and progression of the disease. Both frequency and the amount of consumed carbohydrates are of importance. During the visit, a good practice is also explained to the carers, such as the effects of sugar in acid production and how it affects hard tooth tissues. They should be advised to control the so-called hidden sugar (in syrups and inhalants). In shaping children’s healthy behaviors, the main role is played by the parents, who are the first and primary source of knowledge and skills, hence educating them is a crucial issue.

### 4.2. 6–12 Months

As is recommended by the American Academy of Pediatric Dentistry (AAPD) [29] and the American Dental Association (ADA) [30], a child should see the dentist when the first milk tooth has erupted, but not later than before 12 months of age. Strong recommendations during this visit are to control if hygiene procedures performed by the parents are effective, to assess the oral growth and development and the risk of caries, based on the physical examination and history and to diagnose any pathological conditions (for instance foci of caries). Dietary advice and oral hygiene instruction are also conditionally recommended, including hygienic means suitable for the patient’s age (such as proper toothpastes).

### 4.3. 12–24 Months

Between 12 and 24 months of a child’s life, regular check-ups are necessary, at least every 6 months, depending on recommendations [27]. During the visits, it is strongly recommended to monitor the effectiveness of hygienic procedures, assess the state of dentition (development stage, morphological structure and possible diseases of hard dental tissues) and observe the growth and development of the masticatory system. The patient’s risk of caries should also be assessed, based on their medical history and physical examination. This individual examination is to consider if a new caries lesion is possible in the future or whether existing lesions will progress. A good practice is also to perform professional cleaning procedures together with oral hygiene instruction. Strongly recommended is also to brush the children’s teeth by the parents or guardians, feed the child with non-cariogenic products, limit the consumption of sugar-containing products, introduce a cup to drink starting at the age of 6 months and stop bottle-feeding from about 1 year of age. The topical application of fluoride preparations is also valuable and conditionally recommended. During this period, a good practice is to draw the parents’ attention to the etiology of masticatory system dysfunction and parafunction as well as to the ways to eliminate abnormal habits of that type in the child.

### 4.4. 2–6 Years

In children between 2 and 6 years of age, in addition to the activities listed for younger children, it is strongly recommended to implement check-up visit orthodontic diagnostics. Early orthodontic prophylaxis can eliminate the need for the subsequent treatment of malocclusion. When assessing the risk of tooth decay, it should be stressed that caries lesions found in children under 3 years of age require these children to be assigned to a caries high-risk group. Therefore, they have to undergo appropriate preventive measures.

### 4.5. 6–12 Years

In CF children between 6 and 12 years of age, it is recommended to raise their health awareness. Oral hygiene instruction, including brushing and flossing methods, is strongly recommended to be made available to both patients and their parents. Parents are initially more involved themselves, but over time, it is the growing patients who are becoming responsible for their own health and their potential allowing. When permanent teeth appear, their fissures should be secured with fissure sealants, as recommended.

### 4.6. 12–18 Years

In this age group, apart from the procedures described above, a strong recommendation is to evaluate the level of mouth hydration, basing it on their medical history and physical examination. If CF patients complain of dry mouth, or when examination indicates lower hydration of the oral mucosa (or the symptoms suggestive of it such as crusting of the mouth corners or dry lips) [31], stimulated saliva secretion should be measured using a paraffin block. Among additional investigations, the evaluation of saliva pH or its microbial analysis may be conditionally recommended.

In reduced salivation or even xerostomia the applied treatment is symptomatic. An adequate supply of fluids, especially water, is of the utmost importance. Water should be sipped slowly throughout the day. Relief is also brought about by a proper diet, with no spicy or salty foods. Moreover, salivation may be activated by physical stimulation, for instance by chewing sugar-free gum [31]. In cases of considerable dryness, moisturizing agents might be useful (they contain carboxycellulose or mucin) presented as tablets, sprays, gels or mouthwashes. The feeling of dryness is also effectively relieved by the so-called “artificial saliva”, which covers the oral mucosa with a moist, protective layer. The formula of preparations containing mucin, carboxymethylcellulose, glycoproteins, glycerin and some enzymes [31,32] show similarity to natural saliva composition.

With an adequate amount of saliva lacking, bacterial biofilm stagnation can be expected due to faulty oral self-cleansing. The biofilm will stay not only on the surfaces of the teeth but also on the mucous membranes. Xerostomia may also promote aphthae, mucosal ulceration, bacterial or fungal infections and halitosis [31,32]. In these patients, who are at high risk of oral diseases, there is a strong recommendation that control dental visits will be made more frequent, not only to control oral health and to monitor the success of the bacterial biofilm removal procedures from the teeth, tongue and cheeks, but also to encourage patients to actively participate in treatment, both through the use of appropriate measures and suitable symptom-reducing habits.

Ignoring dry mouth may significantly impact the health and quality of life of CF patients complaining of this problem. Patients should be informed that saliva contains a number of factors involved in the non-specific response, e.g., histatin, defensin or lysozyme. Thus, less saliva means reduced local resistance to infections, both bacterial and fungal. Moreover, inadequate saliva flow and lowering its pH make favorable conditions for the pathogens responsible for caries or periodontal diseases to develop [32].

## 5. Conclusions

Regardless of the CF patient’s age, dental check-ups should be scheduled at least every 6 months. However, taking into account the actual condition of the individual CF patients, therapeutic visits may be scheduled for an earlier date to provide well-fitting treatment, considering the level of risk of oral diseases.

Seemingly, the above-described management standards will be helpful and will improve the quality of dental care provided to CF patients.

## Figures and Tables

**Table 1 jcm-11-02756-t001:** Recommended dental schedule for cystic fibrosis (CF) patients (min. every 6 months or as indicated).

	At the Time of CF Diagnosis	6–12 Months	12–24 Months	2–6 Years	6–12 Years	12 Years and Older	18 Years and Older
an explanation of the possible oral risks associated with cf	●	●	●	●	●	●	●
assessing the systemic and topical fluoride status and providing counseling regarding fluoride	●	●	●	●	●	●	●
parents’ oral cavity sanitation	●	●	●	●			
oral hygiene instruction	● Parents	● Parents	● Parents	● Parents	● Parents/ Patient	● Patient	● Patient
dietary counseling	●	●	●	●	●	●	●
clinical oral examination		●	●	●	●	●	●
assessment of oral growth and development		●	●	●	●	●	
caries risk assessment		●	●	●	●	●	●
counseling for nonnutrive habits		●	●	●	●	●	●
age-appropriate injury prevention counseling for orofacial trauma		●	●	●	●	●	●
removing supra- and subgingival stains or deposits		●	●	●	●	●	●
providing required treatment		●	●	●	●	●	●
prophylaxis and topical fluoride		●	●	●	●	●	●
assessment and treatment of developing malocclusion			●	●	●	●	●
pit and fissure sealants for caries susceptible teeth					●	●	
assessment of oral moisture degree					●	●	●
dry mouth treatment counseling					●	●	●
determine the interval for periodic reevaluation	●	●	●	●	●	●	●

## Data Availability

Not applicable.

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
