# Peer review of "Oral Care Recommendation for Cystic Fibrosis Patients–Recommendation for Dentists"

_jcm, 2022, doi:10.3390/jcm11102756_

Round 1

Reviewer 1 Report

INTRODUCTION

  • Page 1, line number 27: Abbreviations are to be expanded
  • Page 2, line number 50: Grammatical errors are highlighted and need to be revised
  • Page 2, line number 57,84: Spelling error is highlighted and needs to be corrected
  • Page 3, line number 140: Grammatical errors are highlighted and need to be corrected and reframed.

RECOMMENDATIONS

  • Page 4, line number 182, 191: The full stops need to be removed in the side headings and corrected
  • Page 5, line number 208, 215, 222: The full stops need to be removed in the side headings and corrected
  • Page 5, line number 200, 217: Grammatical errors are highlighted and need to be revised
  • Page 6, line number 254: References to be quoted at the end of the sentence.
  • Page 6: table 1: Spelling error in table column needs to be corrected

Author Response

Reviewer #1

We would like to thank the reviewer for the valuable comments and suggestions, which help to improve the quality of this manuscript. The following are our point-by-point responses (the reviewer’s comments are in italics):

INTRODUCTION

  • Page 1, line number 27: Abbreviations are to be expanded

Response:

Thank you very much for this comment. We have expended the abbreviations:

This channel allows not only for the transport of chloride (Cl-) ions, but also of bicarbonate (HCO3-), sodium (Na+), potassium (K+) and calcium (Ca2+) ions as well as water (H2O) through the cell membrane.

  • Page 2, line number 50: Grammatical errors are highlighted and need to be revised
  • Page 2, line number 57,84: Spelling error is highlighted and needs to be corrected done
  • Page 3, line number 140: Grammatical errors are highlighted and need to be corrected and reframed.

Response:

We would like to apologize for the mistakes. The corrections have been made as suggested.

RECOMMENDATIONS

  • Page 4, line number 182, 191: The full stops need to be removed in the side headings and corrected
  • Page 5, line number 208, 215, 222: The full stops need to be removed in the side headings and corrected
  • Page 5, line number 200, 217: Grammatical errors are highlighted and need to be revised
  • Page 6, line number 254: References to be quoted at the end of the sentence.
  • Page 6: table 1: Spelling error in table column needs to be corrected

Response:

Thank you very much for pointing out these mistakes. Your suggestion is very valuable. The correction has been made as suggested by the reviewer:

Reviewer 2 Report

The topic is well presented and interesting for a specific group of patients which is honestly difficult to follow in everyday work experience with standardized protocols.

Authors offer valid recommendations: at this proposal, I suggest to cancel from the title "guidelines" and maintain "recommendations", as the topic is currently not deeply investigated.

I suggest to reduce a bit some paragraphs in the introduction, to make the text more concentrated on clinical aspects and easier to follow.

I suggest to add something about the strength of recommendations described for every age interval proposed by authors.

Author Response

Reviewer #2

We sincerely thank the reviewer for constructive criticisms and comments, which were of great help in revising the manuscript. Please, find below our responses (the reviewer’s comments are in italics).

The topic is well presented and interesting for a specific group of patients which is honestly difficult to follow in everyday work experience with standardized protocols.

Authors offer valid recommendations: at this proposal, I suggest to cancel from the title "guidelines" and maintain "recommendations", as the topic is currently not deeply investigated.

Response:

We appreciate the positive feedback from the reviewer. As suggested by the reviewer, we have changed the title.

I suggest to reduce a bit some paragraphs in the introduction, to make the text more concentrated on clinical aspects and easier to follow.

Response:

Thank you very much for this comment. In the introduction section, we wanted to explain pathomechanism of the abnormalities in CF patients, familiarize dentists with the background of dental problems and then focus on the clinical aspect of prophylaxis and treatment needs of this group of special dental care patients. As suggested we have reduced some paragraphs in the introduction section.

I suggest to add something about the strength of recommendations described for every age interval proposed by authors.

Response:

Thank you very much for pointing out these problems. Your suggestions are very valuable. As suggested by the reviewer, we have revised the manuscript and we have rewritten the recommendations adding their strength of them in the particular aged groups.

This manuscript is a resubmission of an earlier submission. The following is a list of the peer review reports and author responses from that submission.